# How are trial outcomes prioritised by stakeholders from different regions? Analysis of an international Delphi survey to develop a core outcome set in gastric cancer surgery

Bilal Alkhaffaf[1,2]*, Aleksandra Metryka[3], Jane M. Blazeby[4], Anne-Marie Glenny[5], Paula R. Williamson[6‡], Iain A. Bruce[3,7‡], on behalf of the GASTROS International Working Group[¶]

1 Department of Oesophago-Gastric Surgery, Salford Royal Hospital, Salford Royal NHS Foundation Trust, Manchester, United Kingdom, 2 Division of Cancer Sciences, School of Medical Sciences, Faculty of Biology, Medicine and Health, University of Manchester, Manchester, United Kingdom, 3 Paediatric ENT Department, Royal Manchester Children's Hospital, Manchester University NHS Foundation Trust, Manchester, United Kingdom, 4 Centre for Surgical Research and Bristol NIHR Biomedical Research Centre, University of Bristol, Bristol, United Kingdom, 5 Division of Dentistry, School of Medical Sciences, Faculty of Biology, Medicine and Health, University of Manchester, Manchester, United Kingdom, 6 MRC North West Hub for Trials Methodology Research, University of Liverpool and a Member of Liverpool Health Partners, Liverpool, United Kingdom, 7 Division of Infection, Immunity and Respiratory Medicine, Faculty of Biology, Medicine and Health, University of Manchester, Manchester, United Kingdom

‡ PRW and IAB are joint senior authors on this work.
¶ Membership of the GASTROS working group is listed in the Acknowledgments.
* bilal.alkhaffaf@srft.nhs.uk

## Abstract

### Background

International stakeholder participation is important in the development of core outcome sets (COS). Stakeholders from varying regions may value health outcomes differently. Here, we explore how region, health income and participant characteristics influence prioritisation of outcomes during development of a COS for gastric cancer surgery trials (the GASTROS study).

### Methods

952 participants from 55 countries participating in a Delphi survey during COS development were eligible for inclusion. Recruits were grouped according to region (East or West), country income classification (high and low-to-middle income) and other characteristics (e.g. patients; age, sex, time since surgery, mode of treatment, surgical approach and healthcare professionals; clinical experience). Groups were compared with respect to how they categorised 56 outcomes identified as potentially important to include in the final COS ('consensus in', 'consensus out', 'no consensus'). Outcomes categorised as 'consensus in' or 'consensus out' by all 3 stakeholder groups would be automatically included in or excluded from the COS respectively.

**Data Availability Statement:** All relevant data are within the paper and its Supporting Information files.

**Funding:** This study is funded by the National Institute for Health Research (NIHR - www.nihr.ac. uk) Doctoral Research Fellowship Grant (DRF-2015-08-023). The award was received by BA. This paper presents independent research funded by the National Institute for Health Research (NIHR). The views expressed are those of the author(s) and not necessarily those of the NHS, the NIHR or the Department of Health. The funders had no role in study design, data collection and analysis, decision to publish, or preparation of the manuscript.

**Competing interests:** The authors have declared that no competing interests exist.

## Results

In total, 13 outcomes were categorised 'consensus in' (disease-free survival, disease-specific survival, surgery-related death, recurrence of cancer, completeness of tumour removal, overall quality of life, nutritional effects, all-cause complications, intraoperative complications, anaesthetic complications, anastomotic complications, multiple organ failure, and bleeding), 13 'consensus out' and 31 'no consensus'. There was little variation in prioritisation of outcomes by stakeholders from Eastern or Western countries and high or low-to-middle income countries. There was little variation in outcome prioritisation within either health professional or patient groups.

## Conclusion

Our study suggests that there is little variation in opinion within stakeholder groups when participant region and other characteristics are considered. This finding may help COS developers when designing their Delphi surveys and recruitment strategies. Further work across other clinical fields is needed before broad recommendations can be made.

## 1. Introduction

A core outcome set (COS) is an agreed minimum group of critically important outcomes which should be reported by all trials within a research field [1]. The GASTROS study (www. gastrosstudy.org) aims to develop a COS in the field of gastric cancer surgery to promote uniform reporting of important outcomes and facilitate evidence synthesis [2]. This is necessary as there is significant variation and heterogeneity in this field with respect to reporting and measurement of outcomes [3]. Furthermore, the outcomes chosen by researchers to report in surgical trials for gastric cancer often do not reflect the priorities held by patients [4]. For this reason, the GASTROS study has sought consensus between patients and healthcare professionals with respect to outcome selection.

Delphi surveys and consensus meetings are commonly used methodologies in the development of COS [1, 5]. Delphi surveys ask participants deemed by the study group to hold an important perspective (key stakeholders) to prioritise outcomes and achieve consensus. The completed Delphi survey often informs and influences discussions during a subsequent consensus meeting, with the aim of resolving uncertainties regarding prioritisation and ratifying the final composition of the COS. Clear recruitment strategies for Delphi surveys are an important consideration. If recruitment does not result in representative stakeholder groups, there is a risk that the results of the Delphi may not be valid [6]. This is particularly important in international COS where significant regional and cultural differences may influence the results ahead of a consensus meeting and, ultimately, the final COS.

Ensuring stakeholder groups are representative can be a challenging task. There is a need to consider many factors including the incidence of the disease, treatment protocols, international variation in healthcare systems and values and socio-economic issues. In the case of curative surgery for gastric cancer it is known that practice varies worldwide (e.g. how surgery is carried out and the extent of resection) and typically surgeons value different outcomes to patients [4]. For example, due to screening programmes, cancers are generally earlier in the Far East where patients tend to be younger with fewer co-morbidities. There is therefore a need to explore these issues to understand how key stakeholders are selected for survey

participation. In the GASTROS study 952 participants were recruited to a Delphi survey (268 were patients, 445 surgeons and 239 nurses) from 55 countries. It was therefore possible to explore how stakeholder characteristics influenced outcome prioritisation.

This study had two main objectives:

1. To describe the characteristics of Delphi participants and explore their possible influence on the prioritisation of outcomes within stakeholder groups.

2. To explore how stakeholders from different regions prioritised outcomes.

## 2. Methods

This was an analysis of registration data supplied by Delphi survey (S1 File) participants as part of the GASTROS study. Both rounds of the survey took place between March and October 2019. Details of the scope, objectives and methodology of the study have been previously described [2–4]. In summary, participants were asked to score outcomes in terms of importance. The results of the Delphi survey informed discussions in a consensus meeting where final recommendations were made regarding which outcomes to include in the COS.

### 2.1. Stakeholder selection and baseline information

The GASTROS study sought to involve key stakeholders–patients, surgeons, and oncology nurses—to identify a COS for surgical trials in gastric cancer. Our guiding principle has been to promote the 'patient voice' as they are the beneficiaries of trials in this field and have all-important 'lived experience'. The patient voice has previously been shown to be under-represented in COS development [7]. Surgeons provide a clinical perspective and the experience of treating large volumes of patients. Oncology nurses were invited to participate given their central roles as care-givers, patient advocates and core members of the clinical team.

Recruitment was achieved by promoting the study at surgical and nursing congresses, social media and through patient groups and charities. The study website (gastrosstudy.org) allowed stakeholders to register their interest ahead of the Delphi survey. Local recruitment of patient healthcare professionals by members of the international working group was also undertaken. Participation in the Delphi survey was open to all interested stakeholders who fulfilled the following criteria:

- Surgeons who had completed their training and routinely treat gastric cancer.

- Oncology nurses with a recognised proportion of their role involved in the care and follow-up of gastric cancer patients.

- Patients who have undergone surgical resection for gastric cancer with the intention of cure.

There is no sample size requirement for Delphi surveys. To be able to demonstrate the enrolment of a broad and representative range of stakeholders, participants were asked to provide the information listed below:

Patients:

- Age

- Sex

- Surgical approach (laparoscopic or open)

- Type of gastrectomy (total or partial)

- Modality of treatment (surgery alone or a combination of surgery and chemotherapy or radiotherapy)

- Time since surgery

  Surgeons:

- Experience (number of gastrectomies undertaken)

  Nurses:

- Experience (years of service)

These datapoints were developed based on information that was likely to be readily known to participants and the expert opinion of the GASTROS study management group (SMG) with respect to important factors that may influence outcomes or perspectives. In the context of patients, different health outcomes, such as complications and survival, may impact their lived experience and ultimately how outcomes are prioritised. Similarly, as clinical experience changes with time, there may be a greater exposure to and therefore appreciation of the impact or importance of longer-term consequences of surgery.

Additionally, all participants were asked to provide their country of residence so that regional differences could be considered. Participants were categorised into 'Eastern' or 'Western' countries (Fig 1) and 'high-income' or 'low- to medium-income countries' as defined by the Organisation for Economic Co-operation and Development's Development Assistance Committee [8]. Eastern countries were defined as those within East Asia, South East Asia, and Eastern Russia, and included China, Japan, South Korea, Thailand, Vietnam, Malaysia, and Singapore [9]. Western countries were defined as those from Western Europe, North America, Australia, and New Zealand [10]. Contrasting between the 'East' and 'West' is of particular importance to gastric cancer given the differences in incidence, pathology, treatment and outcome. It was hypothesised that these differences in approach and survival may influence how stakeholders in these regions prioritise different health outcomes which could be examined further in this study [11, 12]. Similarly, health priorities may be influenced by resource availability as categorised by country income.

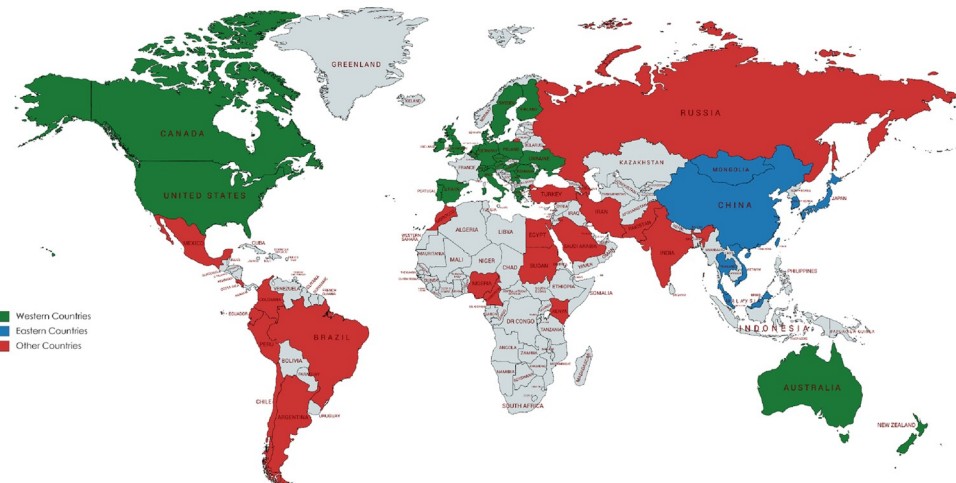

**Fig 1. Countries from which participants were recruited.** Eastern countries were defined as those within East Asia, Southeast Asia, and Eastern Russia, and included China, Japan, South Korea, Thailand, Vietnam, Malaysia, and Singapore [9]. Western countries were defined as those from Western Europe, North America, Australia, and New Zealand [10].

## 2.2 Scoring of outcomes in the Delphi survey and categorisation of outcomes

A list of 56 outcomes identified from previous trials and patient interviews [3, 4] were presented to survey participants who were asked to rate each outcome on a scale of importance (1–3: not important, 4–6: important, 7–9: critically important). Outcomes were organised according to five core areas (mortality/survival; clinical/physiological outcomes; life impact; resource use; adverse events) based on a taxonomy developed for COS development [13]. Patients, surgeons, and nurses group ratings were considered separately to ensure that each group had an equal voice. Participants had the opportunity to suggest further outcomes that they believed had not been presented in round 1. One additional new outcome suggested by participants in round 1 was identified and after consideration by the SMG was presented to participants for scoring in round 2. Therefore, a total of 57 outcomes were presented in round 2 where, for each outcome, participants were shown the scores from each stakeholder group, and given the opportunity to change their rating if they wished.

After two rounds of rating, outcomes were categorised as follows:

- To be included in the COS ('consensus in')

- To be excluded from the COS ('consensus out')

- 'No consensus' reached i.e. no decision reached as to whether the outcome should be included in of excluded from the COS.

Criteria for categorising outcomes was set a priori by the SMG and based on established COS methodology [1]. If an outcome was rated 7–9 (critically important) by 70% or more of a stakeholder group and 1–3 (not important) by no more than 15% of the group, then the consensus amongst that group was that the outcome should be included in the COS. If an outcome was rated 7–9 (critically important) by less than 50% of the group, the consensus amongst that group was for the outcome to be excluded from the COS. Unanimous agreement amongst all three stakeholder groups was required for inclusion in, or exclusion from, the COS. Any other combination resulted in the outcome being placed in the 'no consensus' category and was discussed at a pre-planned consensus meeting to finalise the COS.

## 2.3 Data analysis and interpretation

In round 1, participants completing 50% or more of the Delphi survey were included in the round 1 analysis and invited to participate in round 2. Likewise, participants completing 50% or more of the survey in round 2 were included in the round 2 analysis. For the purpose of this present analysis, participants were placed into 'sub-groups' according to the registration data they submitted (e.g. patient treatment type, surgeon experience etc) to examine the differences in outcome scoring. The following analyses were performed after 2 rounds of ratings:

1. The proportion of participants scoring each outcome as 'critically important' (score 7–9). This analysis approach was chosen as these figures were presented in the consensus meeting discussing results from the Delphi survey.

2. The consensus opinion of each sub-group with respect to whether the outcome should be 'included' in the COS, 'excluded' from the COS or whether 'no consensus' could be reached. These categorisations were compared against the overall 'in', 'out' and 'no consensus' categorisations by each stakeholder group (patients, surgeons and nurses) which was presented to the consensus meeting participants.

Participants not providing demographic data during registration were excluded from the sub-group analyses. When exploring differences in prioritisation, particular focus was placed on outcomes that were categorised as 'consensus in' by one sub-group and 'consensus out' by another.

To examine the possible influence of attrition bias between rounds, the characteristics of stakeholders participating in both rounds were compared to those who only completed round 1. A descriptive analysis was undertaken, and the Chi squared test applied (using SPSS—IBM Corp. Released 2019. IBM SPSS Statistics for Windows, Version 26.0. Armonk, NY: IBM Corp) to examine for statistically significant differences at the 0.05 level.

## 2.4 Ethical approval

The study was given ethical approval by the North West—Greater Manchester East Research Ethics Committee (18/NW/0347) and governance approvals by Manchester University Hospitals NHS Foundation Trust. All participants were provided with a participant information booklet. Informed written consent was obtained during the online registration process for participants in the Delphi survey.

## 3. Results

### 3.1 Overview

The characteristics of participants included in the analysis and attrition rates are summarised in Table 1. After 2 rounds of voting, agreement was reached amongst all three stakeholder groups to include 13 outcomes (disease-free survival, disease-specific survival, surgery-related death, recurrence of cancer, completeness of tumour removal, overall quality of life, nutritional effects, all-cause complications, intraoperative complications, anaesthetic complications, anastomotic complications, multiple organ failure, and bleeding) into the COS. A further 13 outcomes were excluded from the COS (endocrine complications, fatigue, surgical stress response, post-operative psychosis, insomnia, impact on sexual function, ability to eat socially, ability to interact socially, impact on physical appearance, impact on spirituality or faith, wound size, cost and destination on discharge), leaving 31 'no consensus' outcomes for discussion at the consensus meeting.

### 3.2 Prioritisation of outcomes within stakeholder groups (subgroup analysis)

Baseline characteristics reported by stakeholders during the registration process were examined to understand whether these influenced how outcomes were prioritised.

**3.2.1 Patient prioritisation of outcomes.** A summary of outcomes categorised for 'inclusion' into the COS by at least one patient sub-group is presented in Table 2. Thirty outcomes were categorised for inclusion in the COS by at least one subgroup. Four outcomes were simultaneously categorised both for 'inclusion' and 'exclusion' by different subgroups. None of the outcomes categorised for inclusion by all stakeholder groups were voted 'consensus out' by any patient sub-group. Seven outcomes were categorised for inclusion in the COS by all patient subgroups.

**3.2.2 Surgeon prioritisation of outcomes.** Table 3 summarises and compares outcomes categorised for inclusion into the COS by at least one surgeon sub-group. Twenty-one outcomes were categorised for inclusion by at least one subgroup. No outcomes were simultaneously categorised both for 'inclusion' and 'exclusion' by different subgroups. Twelve outcomes were categorised by all surgeon subgroups for inclusion.

**Table 1. Demographic characteristics of participants included in analysis of round 1 and 2 scores.**

| Stakeholder Group | Variable | Sub-Group | Total | Completed round 1 only (%)* | Completed both rounds (%)* | p value |
|---|---|---|---|---|---|---|
| Patients | All | - | 268 | 84 | 184 | |
| | Age | <60 | | 38 (45) | 77 (42) | 0.69 |
| | | > = 60 | | 46 (55) | 107 (58) | |
| | Sex | Male | | 52 (62) | 101 (55) | 0.345 |
| | | Female | | 32 (38) | 83 (45) | |
| | Region | West | | 53 (62) | 113 (74) | 0.461 |
| | | East | | 23 (38) | 39 (26) | |
| | Country income | HIC | | 53 (63) | 113 (61) | 0.792 |
| | | LMIC | | 31 (37) | 71 (39) | |
| | Years since surgery | <1 year | | 15 (19) | 30 (17) | 0.656 |
| | | 1 to 3 years | | 34 (44) | 68 (39) | |
| | | >3 years | | 29 (37) | 75 (43) | |
| | Surgical approach | Open | | 70 (83) | 145 (78) | 0.850 |
| | | MIS | | 14 (17) | 31 (22) | |
| | Type of surgery | Total | | 40 (49) | 78 (44) | 0.503 |
| | | Partial | | 42 (51) | 98 (56) | |
| | Treatment Modality | Surgery alone | | 28 (34) | 69 (39) | 0.495 |
| | | Multimodal therapy | | 54 (66) | 110 (61) | |
| Surgeons | All | - | 445 | 102 | 343 | |
| | Region | West | | 33 (38) | 174 (61) | 0.000 |
| | | East | | 53 (62) | 109 (39) | |
| | Country income | HIC | | 45 (44) | 201 (59) | 0.010 |
| | | LMIC | | 57 (56) | 142 (41) | |
| | Surgeon experience | <50 | | 21 (29) | 70 (23) | 0.45 |
| | | 50–199 | | 20 (27) | 103 (34) | |
| | | >200 | | 32 (44) | 127 (43) | |
| Nurses | All | - | 239 | 104 | 135 | |
| | Region | West | | 22 (35) | 40 (40) | 0.100 |
| | | East | | 57 (65) | 61 (60) | |
| | Country income | HIC | | 24 (23) | 46 (34) | 0.064 |
| | | LMIC | | 80 (77) | 89 (66) | |
| | Nurse experience | 0–5 years | | 59 (57) | 59 (45) | 0.056 |
| | | >5 years | | 44 (43) | 73 (55) | |

HIC = high income country, LMIC = low- to middle-income country; MIS = minimally invasive surgery.

*All percentages refer to the proportion of participants from each sub-group completing either round 1 or both rounds.

**3.2.3 Nurse prioritisation of outcomes.** Table 4 summarises and compares the outcomes categorised for inclusion by at least one nurse sub-group. Twenty-two outcomes were categorised for inclusion by at least one subgroup. Five outcomes were simultaneously categorised both for 'inclusion' and 'exclusion' by different subgroups. None of the outcomes categorised for automatic inclusion by all stakeholder groups were voted 'consensus out' by any nurse subgroup. Ten outcomes were categorised by all nurses' subgroups for inclusion.

## 3.3 Impact of regional variation on prioritisation of outcomes

Table 5 details the final categorisation of outcomes in the Delphi survey as agreed by all stakeholder groups. This is compared to outcome categorisation lists based on the region (East versus

**Table 2. Outcomes categorised for inclusion in the COS by at least one subgroup of patients.**

| Outcome | Overall | Region** | | Country Income | | Age in years | | Sex | | Years since surgery | | | Surgical approach | | Type of gastrectomy | | Treatment Modality | |
|---|---|---|---|---|---|---|---|---|---|---|---|---|---|---|---|---|---|---|
| | All patients | West | East | HIC | LMIC | <60 | >=60 | M | F | <1 | 1–3 | >3 | Open | MIS | Total | Partial | Surgery | Multi-modal |
| | n = 184 | n = 113 | n = 39 | n = 113 | n = 71 | n = 77 | n = 107 | n = 101 | n = 83 | n = 30 | n = 68 | n = 75 | n = 145 | n = 31 | n = 78 | n = 98 | n = 69 | n = 110 |
| **Outcome Area: Mortality/Survival** | | | | | | | | | | | | | | | | | | |
| 1. Disease-free survival* | 85.4 | 87.0 | 76.3 | 87.0 | 82.9 | 86.7 | 84.5 | 83.0 | 88.5 | 86.2 | 89.4 | 86.1 | 85.9 | 85.7 | 84.9 | 85.6 | 80.3 | 87.9 |
| 2. Dying from stomach cancer* | 86.4 | 88.7 | 74.4 | 88.7 | 82.9 | 85.5 | 87.0 | 88.8 | 83.3 | 86.2 | 89.2 | 85.9 | 88.4 | 80.0 | 85.3 | 87.1 | 81.8 | 88.6 |
| 3. Dying from any cause | 66.7 | 65.0 | 77.8 | 65.0 | 69.1 | 72.6 | 62.2 | 59.6 | 75.3 | 77.8 | 63.6 | 66.2 | 67.2 | 63.0 | 65.3 | 66.3 | 71.4 | 62.1 |
| 4. Surgery-related death* | 84.0 | 86.9 | 72.2 | 86.9 | 79.4 | 76.7 | 89.2 | 80.2 | 88.6 | 81.5 | 85.1 | 88.7 | 86.2 | 73.3 | 82.7 | 84.9 | 79.7 | 86.8 |
| **Outcome Area: Clinical/physiological outcomes** | | | | | | | | | | | | | | | | | | |
| 7. Anastomotic complications* | 76.7 | 80.0 | 74.4 | 80.0 | 71.8 | 74.3 | 78.4 | 75.0 | 78.8 | 82.8 | 77.6 | 74.3 | 76.1 | 74.2 | 69.3 | 80.9 | 74.6 | 76.9 |
| 8. Gastro-intestinal functional problems | 72.8 | 85.3 | 71.8 | 85.3 | 53.5 | 69.7 | 75.0 | 66.0 | 81.3 | 75.9 | 63.2 | 80.6 | 70.2 | 77.4 | 70.1 | 72.6 | 67.7 | 74.5 |
| 9. Bowel Complications | 71.8 | 80.0 | 76.9 | 80.0 | 59.2 | 65.8 | 76.2 | 66.0 | 79.0 | 75.9 | 64.7 | 76.7 | 69.7 | 74.2 | 67.5 | 72.9 | 68.2 | 72.7 |
| 12. Multiple organ failure* | 86.4 | 87.9 | 86.5 | 87.9 | 84.1 | 87.8 | 85.3 | 86.7 | 85.9 | 86.2 | 86.4 | 85.7 | 87.8 | 79.3 | 83.8 | 87.2 | 80.3 | 89.5 |
| 16. Hepatic Complications | 62.4 | 65.0 | 73.7 | 65.0 | 57.1 | 71.6 | 54.5 | 52.1 | 73.4 | 78.6 | 52.3 | 60.9 | 61.0 | 62.1 | 60.8 | 59.3 | 67.2 | 56.4 |
| | All patients | West | East | HIC | LMIC | <60 | >=60 | M | F | <1 | 1–3 | >3 | Open | MIS | Total | Partial | Surgery | Multi-modal |
| 17. Pancreatic Complications | 70.3 | 75.5 | 73.7 | 75.5 | 61.4 | 74.0 | 66.7 | 63.8 | 76.9 | 82.1 | 57.8 | 72.5 | 68.9 | 69.0 | 67.1 | 69.2 | 68.7 | 69.0 |
| 18. Abdominal Collection | 71.5 | 72.3 | 82.1 | 72.3 | 70.4 | 71.6 | 71.4 | 65.3 | 79.2 | 75.9 | 70.8 | 67.2 | 69.2 | 77.4 | 63.9 | 75.0 | 73.1 | 69.0 |
| 20. Nutritional Effects* | 73.8 | 77.7 | 69.2 | 77.7 | 66.2 | 75.3 | 71.7 | 69.0 | 78.3 | 73.3 | 72.1 | 75.7 | 72.9 | 71.0 | 75.6 | 70.1 | 69.1 | 74.5 |
| 21. Recurrence of Cancer* | 92.2 | 95.4 | 84.6 | 95.4 | 85.9 | 88.0 | 94.3 | 92.0 | 91.3 | 93.1 | 88.1 | 95.9 | 91.5 | 90.3 | 88.0 | 93.8 | 88.1 | 93.5 |
| 22. Renal complications | 70.0 | 80.0 | 65.8 | 80.0 | 54.3 | 66.2 | 71.7 | 67.0 | 72.4 | 82.1 | 53.8 | 80.3 | 69.2 | 65.5 | 64.8 | 71.4 | 67.2 | 71.4 |
| 23. Urinary complications | 58.1 | 65.7 | 57.9 | 65.7 | 45.7 | 54.2 | 60.0 | 54.2 | 61.8 | 64.3 | 40.0 | 70.6 | 57.8 | 51.7 | 50.0 | 60.9 | 56.1 | 59.4 |
| 25. Respiratory complications | 69.5 | 67.0 | 66.7 | 67.0 | 73.2 | 70.3 | 68.9 | 70.7 | 67.9 | 75.0 | 73.1 | 63.4 | 71.2 | 56.7 | 60.8 | 73.7 | 68.2 | 68.9 |
| 27. Cerebro-vascular complications | 77.6 | 81.0 | 68.4 | 81.0 | 72.9 | 68.6 | 84.0 | 80.9 | 73.7 | 75.0 | 72.3 | 84.8 | 78.0 | 73.3 | 69.0 | 82.4 | 71.2 | 82.8 |
| 28. Thrombo-embolic complications | 76.7 | 80.4 | 63.2 | 80.4 | 71.4 | 73.2 | 79.2 | 78.9 | 74.0 | 71.4 | 73.8 | 82.4 | 79.9 | 60.0 | 73.6 | 77.2 | 66.7 | 84.2 |
| 29. Bleeding* | 72.3 | 67.6 | 76.9 | 67.6 | 78.9 | 77.5 | 68.6 | 67.0 | 78.9 | 75.0 | 72.7 | 69.1 | 73.3 | 66.7 | 70.0 | 71.6 | 72.7 | 70.6 |
| **Outcome Area: Life impact** | | | | | | | | | | | | | | | | | | |
| | All patients | West | East | HIC | LMIC | <60 | >=60 | M | F | <1 | 1–3 | >3 | Open | MIS | Total | Partial | Surgery | Multi-modal |
| 30. Ability to undertake physical activities | 60.4 | 65.8 | 56.4 | 65.8 | 50.7 | 56.6 | 62.3 | 63.0 | 56.1 | 51.7 | 55.9 | 64.9 | 56.6 | 71.0 | 50.6 | 63.9 | 55.2 | 61.8 |
| 36. Impact on mental health | 58.8 | 61.3 | 48.7 | 61.3 | 56.3 | 57.9 | 60.4 | 61.0 | 57.3 | 55.2 | 58.8 | 64.9 | 55.9 | 71.0 | 63.6 | 54.6 | 52.2 | 63.6 |
| 40. Overall quality of life* | 74.0 | 79.1 | 56.4 | 79.1 | 66.2 | 72.4 | 75.2 | 74.0 | 74.1 | 72.4 | 77.9 | 74.0 | 72.5 | 80.6 | 77.9 | 69.8 | 64.2 | 81.7 |
| 42. Ability to complete treatment pathway. | 79.8 | 83.2 | 69.2 | 83.2 | 74.6 | 81.1 | 78.8 | 81.8 | 77.2 | 79.3 | 83.6 | 78.9 | 77.7 | 83.9 | 81.3 | 76.8 | 71.2 | 85.0 |

(Continued)

**Table 2.** (Continued)

| | Overall | Region** | | Country Income | | Age in years | | Sex | | Years since surgery | | | Surgical approach | | Type of gastrectomy | | Treatment Modality | |
|---|---|---|---|---|---|---|---|---|---|---|---|---|---|---|---|---|---|---|
| 43. Completeness of tumour removal* | 92.8 | 95.5 | 87.2 | 95.5 | 88.7 | 90.9 | 94.2 | 93.9 | 91.5 | 93.3 | 91.2 | 97.2 | 91.5 | 96.8 | 92.2 | 92.7 | 88.2 | 95.4 |
| 44. Conversion to open surgery | 51.2 | 53.6 | 81.6 | 53.6 | 47.7 | 52.2 | 50.5 | 43.0 | 60.5 | 73.3 | 31.7 | 58.1 | 48.0 | 62.1 | 48.5 | 50.6 | 59.1 | 42.9 |
| **Outcome Area: Resource use** | | | | | | | | | | | | | | | | | | |
| 53. Duration of stay in an intensive care ward | 64.1 | 54.4 | 62.9 | 54.4 | 77.6 | 59.2 | 66.7 | 60.6 | 67.1 | 57.7 | 71.2 | 56.7 | 65.7 | 46.4 | 64.3 | 59.8 | 62.5 | 63.4 |
| **Outcome Area: Adverse events** | | | | | | | | | | | | | | | | | | |
| 54. Adverse drug reaction | 67.0 | 72.2 | 59.0 | 72.2 | 59.2 | 64.5 | 68.9 | 64.3 | 70.4 | 51.7 | 64.7 | 77.5 | 66.0 | 66.7 | 66.7 | 64.6 | 67.6 | 66.0 |
| 55. All-cause complications* | 75.8 | 76.6 | 71.8 | 76.6 | 74.6 | 76.0 | 75.7 | 70.7 | 82.3 | 67.9 | 79.1 | 77.8 | 77.0 | 71.0 | 77.0 | 72.9 | 77.6 | 73.6 |
| 56. Intra-operative complications* | 80.6 | 82.9 | 79.5 | 82.9 | 77.1 | 76.7 | 83.3 | 77.3 | 84.6 | 79.3 | 77.3 | 84.3 | 80.9 | 80.6 | 76.7 | 81.9 | 80.3 | 79.8 |
| 57. Anaesthetic complications* | 74.9 | 78.1 | 66.7 | 78.1 | 70.0 | 74.0 | 75.5 | 71.4 | 79.2 | 55.2 | 75.8 | 81.4 | 77.2 | 61.3 | 74.0 | 73.4 | 73.8 | 74.3 |

Values are the percentage of participants voting the outcome as critically important (score 7–9).

Green = for inclusion, Yellow = no consensus, Red = for exclusion. HIC = high income country, LMIC = low- to middle-income country; MIS = minimally invasive surgery.

*Denotes outcomes are those which were included in the final list of outcomes for automatic inclusion in the COS.

**Participants not from either Western or Eastern countries were excluded from this analysis.

**Table 3. Outcomes categorised for inclusion in the COS by at least one subgroup of surgeons.**

| | Overall | Region** | | Country Income | | Cases performed | | |
|---|---|---|---|---|---|---|---|---|
| | All surgeons | West | East | HIC | LMIC | <50 | 50–199 | >200 |
| Outcome | n = 343 | n = 174 | n = 109 | n = 201 | n = 142 | n = 70 | n = 103 | n = 127 |
| **Outcome Area: Mortality/Survival** | | | | | | | | |
| 1. Disease-free survival* | 97.7 | 97.7 | 98.1 | 98.0 | 97.2 | 95.7 | 99.0 | 97.6 |
| 2. Dying from stomach cancer* | 96.5 | 97.7 | 95.4 | 96.0 | 97.2 | 95.7 | 95.1 | 96.9 |
| 4. Surgery-related death* | 96.8 | 96.6 | 99.1 | 97.5 | 95.8 | 94.3 | 96.1 | 98.4 |
| **Outcome Area: Clinical/physiological outcomes** | | | | | | | | |
| 7. Anastomotic complications* | 95.3 | 95.4 | 95.4 | 96.0 | 95.1 | 95.7 | 94.2 | 96.1 |
| 8. Gastro-intestinal functional problems | 74.9 | 75.3 | 70.6 | 75.1 | 76.1 | 82.9 | 76.7 | 67.7 |
| 12. Multiple organ failure* | 81.3 | 81.0 | 78.9 | 81.1 | 80.9 | 75.7 | 83.5 | 81.7 |
| 18. Abdominal Collection | 73.4 | 75.1 | 67.0 | 74.5 | 73.2 | 71.4 | 69.9 | 78.7 |
| 20. Nutritional Effects* | 72.8 | 74.6 | 66.1 | 73.5 | 73.9 | 77.1 | 75.7 | 69.3 |
| 21. Recurrence of Cancer* | 97.7 | 99.4 | 95.4 | 99.0 | 96.5 | 97.1 | 100.0 | 97.6 |
| 25. Respiratory complications | 66.5 | 70.1 | 59.6 | 70.6 | 62.0 | 65.7 | 67.0 | 70.1 |
| 28. Thrombo-embolic complications | 64.1 | 63.2 | 60.6 | 63.2 | 65.5 | 61.4 | 59.2 | 70.9 |
| 29. Bleeding* | 87.5 | 84.5 | 95.4 | 86.1 | 90.1 | 81.4 | 85.4 | 92.9 |
| **Outcome Area: Life impact** | | | | | | | | |
| 30. Ability to undertake physical activities | 66.4 | 71.8 | 59.6 | 69.7 | 63.4 | 65.7 | 70.9 | 66.9 |
| 40. Overall quality of life* | 86.5 | 93.1 | 75.9 | 90.0 | 82.3 | 91.4 | 87.4 | 85.7 |
| 42. Ability to complete treatment pathway. | 78.6 | 86.2 | 61.1 | 82.6 | 73.6 | 87.0 | 74.8 | 75.4 |
| 43. Completeness of tumour removal* | 97.4 | 98.3 | 97.2 | 98.5 | 95.7 | 91.4 | 99.0 | 99.2 |
| **Outcome Area: Resource use** | | | | | | | | |
| 49. Readmission to hospital | 78.9 | 78.7 | 82.4 | 78.6 | 80.9 | 80.0 | 81.6 | 81.0 |
| 51. Need for an additional intervention. | 75.4 | 82.8 | 59.3 | 81.6 | 66.7 | 78.6 | 78.6 | 71.4 |
| **Outcome Area: Adverse events** | | | | | | | | |
| 55. All-cause complications* | 81.2 | 81.5 | 84.3 | 83.0 | 79.4 | 81.4 | 76.7 | 88.1 |
| 56. Intra-operative complications* | 91.5 | 88.4 | 93.5 | 89.5 | 93.6 | 91.4 | 92.2 | 92.9 |
| 57. Anaesthetic complications* | 70.5 | 70.3 | 71.0 | 70.4 | 70.7 | 68.6 | 66.0 | 75.2 |

Values are the percentage of participants voting the outcome as critically important (score 7–9).

Green = for inclusion, Yellow = no consensus. HIC = high income country, LMIC = low- to middle-income country;

*Denotes outcomes are those which were included in the final list of outcomes for automatic inclusion in the COS.

**Participants not from either Western or Eastern countries were excluded from this analysis.

West) and country income differences (HIC versus LMIC). Consensus agreement to include 8 and exclude 7 outcomes was reached across all regional sub-groups. No outcomes were simultaneously categorised as 'consensus in' and 'consensus out' across different regional sub-groups.

## 4. Discussion

The GASTROS study (www.gastrosstudy.org) is the first to bring together healthcare professionals and patients with the purpose of identifying outcomes to include in a COS for surgical

**Table 4. Outcomes categorised for inclusion in the COS by at least one subgroup of nurses.**

| | Overall | Region** | | Country Income | | Experience in years | |
|---|---|---|---|---|---|---|---|
| | All nurses | West | East | HIC | LMIC | 0–5 years | >5 |
| Outcome | n = 135 | n = 40 | n = 61 | n = 46 | n = 89 | n = 59 | n = 73 |
| **Outcome Area: Mortality/Survival** | | | | | | | |
| 1. Disease-free survival* | 85.1 | 92.5 | 85.2 | 93.5 | 80.9 | 81.4 | 89.0 |
| 2. Dying from stomach cancer* | 80.0 | 90.0 | 72.1 | 91.3 | 74.2 | 74.6 | 83.6 |
| 3. Dying from any cause | 63.4 | 64.1 | 70.5 | 64.4 | 65.2 | 58.6 | 71.2 |
| 4. Surgery-related death | 77.6 | 95.0 | 65.6 | 93.5 | 69.3 | 72.9 | 81.9 |
| **Outcome Area: Clinical/physiological outcomes** | | | | | | | |
| 7. Anastomotic complications* | 84.4 | 97.5 | 82.0 | 97.8 | 76.4 | 79.7 | 89.0 |
| 8. Gastro-intestinal functional problems | 69.6 | 90.0 | 65.6 | 89.1 | 57.3 | 59.3 | 75.3 |
| 12. Multiple organ failure* | 79.9 | 82.5 | 78.3 | 84.8 | 78.4 | 83.1 | 79.2 |
| 13. Pain | 59.3 | 85.0 | 59.0 | 87.0 | 44.9 | 49.2 | 65.8 |
| 18. Abdominal Collection | 65.9 | 65.0 | 67.2 | 69.6 | 61.8 | 49.2 | 76.7 |
| 19. Other infections | 61.2 | 55.0 | 70.0 | 58.7 | 61.4 | 54.2 | 65.3 |
| 20. Nutritional Effects* | 74.8 | 87.5 | 77.0 | 87.0 | 66.3 | 69.5 | 76.7 |
| 21. Recurrence of Cancer* | 88.0 | 97.5 | 86.9 | 97.8 | 82.8 | 84.5 | 90.3 |
| 26. Wound complications | 67.4 | 62.5 | 73.8 | 63.0 | 67.4 | 67.8 | 64.4 |
| 29. Bleeding* | 80.7 | 72.5 | 85.2 | 76.1 | 82.0 | 79.7 | 80.8 |
| **Outcome Area: Life impact** | | | | | | | |
| 30. Ability to undertake physical activities | 56.3 | 72.5 | 54.1 | 73.9 | 46.1 | 54.2 | 56.2 |
| 36. Impact on mental health | 54.5 | 70.0 | 48.3 | 71.7 | 44.3 | 54.2 | 52.8 |
| 40. Overall quality of life* | 70.4 | 90.0 | 67.2 | 89.1 | 59.6 | 61.0 | 76.7 |
| 42. Ability to complete treatment pathway. | 65.9 | 77.5 | 60.7 | 78.3 | 58.4 | 54.2 | 75.3 |
| 43. Completeness of tumour removal* | 87.3 | 100.0 | 86.9 | 97.8 | 82.0 | 83.1 | 91.8 |
| **Outcome Area: Resource use** | | | | | | | |
| 49. Readmission to hospital | 69.9 | 77.5 | 68.3 | 78.3 | 62.1 | 60.3 | 73.6 |
| 51. Need for an additional intervention. | 56.7 | 75.0 | 48.3 | 76.1 | 45.5 | 44.1 | 63.9 |
| 52. Need for pain relief | 68.4 | 72.5 | 72.9 | 73.9 | 63.2 | 57.6 | 74.6 |
| **Outcome Area: Adverse events** | | | | | | | |
| 55. All-cause complications* | 77.9 | 77.5 | 77.2 | 80.4 | 75.3 | 70.2 | 83.1 |
| 56. Intra-operative complications* | 85.4 | 90.0 | 91.1 | 91.3 | 83.3 | 85.7 | 87.3 |
| 57. Anaesthetic complications* | 78.0 | 80.0 | 77.8 | 80.4 | 76.5 | 70.9 | 84.1 |

Values are the percentage of participants voting the outcome as critically important (score 7–9). Green = for inclusion, Red = for exclusion, Yellow = no consensus. HIC = high income country, LMIC = low- to middle-income country;

*Denotes outcomes are those which were included in the final list of outcomes for automatic inclusion in the COS.

**Participants not from either Western or Eastern countries were excluded from this analysis.

trials in gastric cancer. The multi-language survey recruited a broad spectrum of stakeholders with different personal and professional experiences from over 50 countries across 6 continents. We aimed to examine whether certain stakeholder characteristics influenced how outcomes were prioritised and whether there were regional influences also. Our analysis from nearly 1000 survey participants suggested that little variation within the stakeholder groups exists. Similarly, when all stakeholders were categorised according to region or country income, significant differences were not identified. These are important findings which should

**Table 5. Regional differences in consensus on outcomes voted for inclusion or exclusion from the COS by at least 1 subgroup.**

| Consensus outcome | Final list of outcomes as agreed by all stakeholder groups | Region** | | Country Consensus income | |
|---|---|---|---|---|---|
| | | West (n = 327) | East (n = 209) | HIC (n = 360) | LMIC (n = 302) |
| **Outcome Area: Mortality/Survival** | | | | | |
| 1. Disease-free survival* | Consensus in | Consensus in | Consensus in | Consensus in | Consensus in |
| 2. Dying from stomach cancer* | Consensus in | Consensus in | Consensus in | Consensus in | Consensus in |
| 4. Surgery-related death | Consensus in | Consensus in | No consensus | Consensus in | No consensus |
| **Outcome Area: Clinical/physiological outcomes** | | | | | |
| 6. Endocrine complications | Consensus out | Consensus out | No consensus | Consensus out | Consensus out |
| 7. Anastomotic complications* | Consensus in | Consensus in | Consensus in | Consensus in | Consensus in |
| 8. Gastro-Intestinal functional problems | No consensus | Consensus in | No consensus | Consensus in | No consensus |
| 11. Fatigue | Consensus out | Consensus out | Consensus out | Consensus out | Consensus out |
| 12. Multiple organ failure* | Consensus in | Consensus in | Consensus in | Consensus in | Consensus in |
| 14. Surgical Stress Response | Consensus out | Consensus out | No consensus | Consensus out | No consensus |
| 15. Gallbladder complications | No consensus | No consensus | No consensus | No consensus | Consensus out |
| 20. Nutritional Effects | Consensus in | Consensus in | No consensus | Consensus in | No consensus |
| 21. Recurrence of Cancer* | Consensus in | Consensus in | Consensus in | Consensus in | Consensus in |
| 23. Urinary complications | No consensus | No consensus | No consensus | No consensus | Consensus out |
| 24. Post-operative psychosis | Consensus out | Consensus out | Consensus out | Consensus out | Consensus out |
| 29. Bleeding | Consensus in | No consensus | Consensus in | No consensus | Consensus in |
| 31. Insomnia | Consensus out | Consensus out | No consensus | Consensus out | Consensus out |
| **Outcome Area: Life impact** | | | | | |
| 32. Impact on sexual function | Consensus out | Consensus out | Consensus out | Consensus out | Consensus out |
| 33. Ability to eat socially | Consensus out | No consensus | Consensus out | No consensus | Consensus out |
| 34. Ability to Interact socially | Consensus out | No consensus | Consensus out | No consensus | Consensus out |
| 35. Impact of surgery on social and work roles | No consensus | No consensus | Consensus out | No consensus | Consensus out |
| 36. Impact on mental health | No consensus | No consensus | Consensus out | No consensus | No consensus |
| 37. Impact on Physical Appearance | Consensus out | Consensus out | Consensus out | Consensus out | Consensus out |
| 39. Impact on spirituality or faith | Consensus out | Consensus out | Consensus out | Consensus out | Consensus out |

*(Continued)*

**Table 5.** (Continued)

| Consensus outcome | Final list of outcomes as agreed by all stakeholder groups | Region** | | Country Consensus income | |
| --- | --- | --- | --- | --- | --- |
| | | West (n = 327) | East (n = 209) | HIC (n = 360) | LMIC (n = 302) |
| 40. Overall quality of life | Consensus in | Consensus in | No consensus | Consensus in | No consensus |
| 41. Impact on perception of physical health | No consensus | No consensus | No consensus | No consensus | Consensus out |
| 42. Ability to complete treatment pathway. | No consensus | Consensus in | No consensus | Consensus in | No consensus |
| 43. Completeness of tumour removal* | Consensus in | Consensus in | Consensus in | Consensus in | Consensus in |
| 45. Duration of surgery | No consensus | Consensus out | No consensus | Consensus out | No consensus |
| 46. Wound size | Consensus out | Consensus out | Consensus out | Consensus out | Consensus out |
| **Outcome Area: Resource use** | | | | | |
| 47. Cost | Consensus out | Consensus out | No consensus | Consensus out | No consensus |
| 50. Destination on Discharge | Consensus out | Consensus out | Consensus out | Consensus out | Consensus out |
| **Outcome Area: Adverse events** | | | | | |
| 55. All-cause complications* | Consensus in | Consensus in | Consensus in | Consensus in | Consensus in |
| 56. Intra-operative complications* | Consensus in | Consensus in | Consensus in | Consensus in | Consensus in |
| 57. Anaesthetic complications | Consensus in | Consensus in | No consensus | Consensus in | Consensus in |

Green = for inclusion, Red = for exclusion, Yellow = no consensus. HIC = high income country, LMIC = low- to middle-income country;

*Denotes outcome was categorised as for 'inclusion' in COS by all subgroups.

**Participants not from either Western or Eastern countries were excluded from this analysis.

serve to reassure researchers and patients that the resulting COS has sought and considered international opinion which is widely representative. Furthermore, these findings suggest that priorities within stakeholder groups and across regions are more aligned than may have been previously thought.

## 4.1 Planning recruitment to Delphi surveys

Few studies have previously examined factors which influence how stakeholders prioritise outcomes in the field of COS development. The BRAVO study explored this in the field of breast cancer reconstruction and found that priorities varied within patient and healthcare professional groups [6]. This led them to recommend careful participant selection for Delphi surveys by COS developers. These same differences, however, were not identified in our study. The BRAVO study's healthcare professional stakeholder group was more heterogenous than the groups in this study (breast surgeons, plastic surgeons, nurses and psychologists grouped together) and so these differences may be expected. Furthermore, reconstructive breast surgery is a complex area which covers many different types of procedures. This may also account for the significant variation in outcome prioritisation by patients which was not mirrored in the GASTROS study. In comparison, gastric cancer surgery generally comprises of either a partial

or total gastrectomy both of which can result in similar short and long-term problems which may explain why priorities were more aligned within stakeholder groups. Similarly, a COS study in the field of bariatric surgery identified significant variation in outcome prioritisation amongst healthcare professionals [14]. Again, healthcare professionals in this study were heterogenous, which supports our strategy to separate surgeons and nurses into different stakeholder groups.

Achieving the 'correct balance' of representative stakeholders is an important consideration during the design phase. For example, knowledge of the patient demographic and which types of interventions are prevalent within that group, will enable researchers to recruit an appropriate number of stakeholders with those characteristics. As this is the first study to specifically examine regional variations amongst stakeholders in COS development, it is unknown whether these findings necessary mirror those from other COS studies. Further examination of previously undertaken Delphi surveys is required, and future surveys will need to ensure that similar baseline characteristics are recorded. This is relatively straightforward information to capture and can be supplied quickly and easily by survey participants during the registration process.

With respect to the GASTROS study, the importance of seeking international agreement on core outcomes was identified at the conception stage and subsequently influenced the design of the prioritisation exercise. Our strategy for addressing the significant challenges associated with international involvement included 1) an international working group with regional collaborators, 2) translating surveys and 3) seeking the support of relevant patient and professional groups. Transparent reporting of methodological approaches adopted during COS development are of paramount importance. Ultimately, a COS will only achieve its stated goals if researchers use it. And whilst there are likely several factors which influence the uptake of COS, ensuring researchers have the confidence that the COS is relevant to them and has been developed through a methodologically robust process are likely to be important factors which influence uptake and dissemination [15].

There are challenges in deciding how to sample participants for a Delphi study. Epidemiological studies, registries and audits provide descriptive regional or national information [16–18]. However, in the case of gastric cancer, these resources are not always complete or available. Consequently, the study team widened the promotion and enrolment into the Delphi to capture as many patients as possible. In our study, we demonstrated that there was not significant variation in outcome prioritisation within stakeholder sub-groups with respect to the characteristics that we examined. Consequently, whilst over 1000 participants were enrolled, it may not have been necessary to recruit such large numbers. This will likely guide our recruitment strategy during future planned stages of work when reviewing the COS and identifying outcome measurement instruments. Our experience may also help guide other COS developers as they consider the number of participants to recruit to their Delphi surveys. However, given some of our findings differed from those in the field of breast surgery reconstruction and bariatric surgery, more work is needed before broad recommendations can be made.

## 4.2 Variations within stakeholder groups

When regional variations across the three stakeholder groups were compared, the greatest differences in prioritisation were observed amongst nurses. For example, in four outcomes (pain, ability to undertake physical exercise, impact on mental health, need for additional intervention) different subgroups of nurses categorised them as 'consensus in' and 'consensus out'. These outcomes seemed less important in LMIC and HIC settings within the nurse group. Understanding the reason for this is likely to be complex. It may be argued that this is simply

because nurses are reflecting the importance that patients from these cultures or regions place on these outcomes as similar trends were seen amongst patients. Limited resource in LMIC settings which may affect follow-up may also play a role in understanding how important longer-term problems are in these regions. Further exploration using qualitative research methods may help understand these differences further.

In examining the differences between patient sub-groups, one would expect to see some differences given the number of characteristics that were examined. Despite this, only two outcomes (urinary complications and conversion to open surgery) were simultaneously categorised as 'consensus in' and 'consensus out' by different sub-groups. This finding suggests that despite the many possible influences on patient experience following gastric cancer surgery, there is not a significant variation in how health related outcomes are prioritised in this group. Surgeons had the greatest concordance with respect to outcome prioritisation. Overall, the observed differences in outcome prioritisation were small within each stakeholder group reassuring researchers using the COS that it is based on the views of a representative cohort of patients and healthcare professionals.

## 4.3 Impact of regional variations on outcomes automatically included in COS

The aim of a COS is to identify outcomes which are critically important across all stakeholder groups participating in the process. In the case of the GASTROS study, an outcome would only be automatically included in the COS if patients, surgeons, and nurses each categorise it 'consensus in'. Ultimately, it is not possible to confidently assess how regional differences may have affected the final categorisation of outcomes which informed the consensus meeting. Participants in round 2 were shown the scores of all stakeholder groups from round 1 before being asked to change their score if they wish. To assess regional differences, Western participants, for example, in round 2 would have needed to see only Western stakeholder group scores from round 1. Furthermore, there are a number of other confounding factors which influence why participants change scores between rounds (see below) further making an analysis of regional impacts challenging.

Despite this, some assessments could be made. No outcomes categorised for automatic inclusion by all three stakeholder groups were categorised for automatic exclusion by a regional sub-group. And no outcomes categorised for automatic exclusion from the COS by all three stakeholder groups were categorised for automatic inclusion by a regional sub-group. This suggests that the regional differences in approach to management or patient outcome may not significantly influence how stakeholders prioritise outcomes

There were two outcomes (gastrointestinal functional problems and ability to complete treatment pathway) which were categorised for automatic inclusion by stakeholders from the West and HIC that were not included in the final list presented to the consensus meeting. Furthermore, some outcomes (surgery-related death, nutritional outcomes, bleeding, overall quality of life, anaesthetic complications) did not reach consensus for automatic inclusion by regional sub-groups yet were automatically included when the overall views of stakeholders were considered. This may bring some to the conclusion that different COS should be developed for different regions as some researchers may be collecting outcomes that were not deemed critically important in their region. However, researchers should be cognisant of the fact that their trials are internationally relevant and vitally important to the larger picture where evidence synthesis is concerned. From a different perspective, some researchers may feel aggrieved if outcomes which are critically important in their region are not eventually included in the COS. It is important to emphasise that COS are minimum reporting guidelines

and that researchers are encouraged to report additional outcomes that they believe are important.

## 4.4 Dissemination of results

From the study's inception, the management team understood the importance of regularly reporting findings to encourage participation from all stakeholder groups. Regular reporting also aimed to increase the study's exposure and highlight its importance to minimising research waste in future trial design. Finally, uptake of the COS requires researchers and funders to have knowledge of its existence. A clear dissemination policy was set out *a priori* and included scientific publications, presentations at medical and nursing congresses as well as lay summaries delivered to patient participants through patient groups and social media. The success of this policy to this point has been reflected by nearly 1000 participants being recruited to the Delphi survey. Continued efforts will be required to ensure that the COS is used, and researchers and patients benefit from it.

## 4.5 Strengths and limitations

Strengths of this study include that it is novel and that was able to recruit a large number of participants from many countries. However, there are some limitations which should be acknowledged. This was an analysis which was not powered to make definitive conclusions about relationships between sub-groups and how outcomes were rated. Therefore, the results should be viewed in this context. Furthermore, the sub-groups examined in this paper were chosen by members of the study team based on their extensive experience in the field of gastric cancer and their understanding of factors which may impact on stakeholder experience, perceptions and subsequently how outcomes may be prioritised. It is possible that other unexplored characteristics impact on how stakeholders prioritise outcomes. In addition, this study did not explore how different characteristics interact with one another to impact on outcome prioritisation (e.g. years since surgery and type of gastrectomy). Doing so would create results which would remove the focus from regional differences and would be difficult to interpret. Furthermore, there were significantly fewer patients from Eastern countries enrolled compared to their Western counterparts. This may have influenced how outcomes were categorised ahead of the consensus meeting. However, due to the interplay of other factors described above, reaching a definite conclusion about the degree of this possible limitation is difficult. This is an area that may benefit from further exploration using qualitative research methods.

Delphi surveys are an established method of reaching consensus in the design of COS [1]. They give participants the opportunity to reflect on their ratings from previous rounds before giving a final score. Only after this opportunity should all scores be analysed, and outcomes categorised ahead of the consensus meeting. During the process of rating outcomes in round 2 of the survey, participants are shown the results from each separate stakeholder group in round 1. The topic of why participants change their scores between rounds is an interesting one which has been examined elsewhere [19]. Through our previous analysis we identified that the reasons for changing scores provided by stakeholders were varied, including having the time to reflect on the question being asked, changing their minds on the importance, impact or usefulness of the outcome in question, and changes in personal experience of the outcome. In fact, the influence of other stakeholder ratings as a reason for significantly changing a score in round 2 was cited by only a minority of healthcare professionals and patients. Another factor which may influence scores between rounds is attrition. Our strategy to keep this as low as possible, alongside other approaches to facilitate international participation in Delphi surveys for COS is a topic which will be described separately. Whilst overall attrition

was 30%, the group this affected the most were nurses with nearly 45% attrition. However, the characteristics of those completing both rounds were not significantly different to those only completing round 1. Likewise, a statistically significant difference was identified in the characteristics of surgeons completing both rounds who were predominantly Western and from HIC compared to the balance of surgeons completing round 1. It could be argued therefore that retaining a greater number of Eastern and LMIC surgeons may have led to slightly different survey results. However, whilst statistically significant, this difference is unlikely to be clinically significant given that the number of surgeons not participating in round 2 was relatively small.

One may consider that, given the multimodal nature of treatment for gastric cancer, a COS would be more relevant if it incorporated all therapies (e.g. chemotherapy, radiotherapy, and endotherapy) and that the views of oncologists should also be sought. However, at the time that GASTROS was conceived, there were 24 ongoing surgical trials planning to recruit 11 000 patients for whom non-surgical-related outcomes would not be applicable or relevant. Other considerations such as the resource and time required to recruit participants from other stakeholder groups were also important, hence why a pragmatic decision was made to limit stakeholder groups to those chosen in this study.

Our methodology for COS development was based on an established approach as described by the COMET handbook [1] and COS developers. This aims to seek consensus based on identifying a long list of potentially important outcomes from a systematic review and patient interviews, followed by a Delphi survey, concluded by a consensus meeting. Whilst this approach is well-studied and has been adopted by many, our experience indicates that the process, particularly when seeking broad international opinion, can be both time and resource intensive. Some groups have already begun to explore whether COS development can be streamlined [20]. It should therefore be acknowledged that COS methodology is a developing field for which a single 'optimal' approach does not necessarily exist. Examination of differing methodological adaptations should therefore form an important part of future COS development studies and appropriate funding should be made available to support this.

## 4.6 Conclusion

The GASTROS Delphi survey recruited a broad spectrum of international stakeholders of patients, nurses and surgeons to produce a list of outcomes which should be included or excluded from a COS and others which required further discussion at a consensus meeting. Consensus across these groups was achieved to include 13 outcomes into the COS which will be discussed further at a final consensus meeting (disease-free survival, disease-specific survival, surgery-related death, recurrence of cancer, completeness of tumour removal, overall quality of life, nutritional effects, all-cause complications, intraoperative complications, anaesthetic complications, anastomotic complications, multiple organ failure, and bleeding). Whilst some regional differences were highlighted, there was little variation within stakeholder groups and between regions with respect to how outcomes were prioritised. This may reassure COS users that the adopted methodology was robust and that the views captured during its development were representative. COS developers should carefully consider the characteristics of Delphi survey participants when planning their recruitment strategy. These variables should be explored further to examine the generalisability of this study's findings.

## Supporting information

**S1 File. Full version of the Delphi survey which was translated into target language.**
(DOCX)

## Acknowledgments

The authors would like to highlight the role undertaken by Dr Aleksandra Metryka, Senior Clinical Trials Coordinator, who facilitated the running of this study. In addition, the authors would like to thank the following associations and groups for their support in facilitating recruitment to the GASTROS study Delphi survey:

- The International Gastric Cancer Association (www.igca.info)

- The Association of Upper Gastro-Intestinal Surgeons of Great Britain and Ireland (www.augis.org)

- The Brazilian Gastric Cancer Association (www.abcg.org.br)

- The Canadian Gastric Cancer Association (www.gastriccancer.ca)

- The Chinese Gastric Cancer Association

- The Dutch Upper GI Cancer Group (www.ducg.nl)

- The GASTRODATA group (www.gastrodata.org)

- Italian Research Group for Gastric Cancer (www.gircg.it)

- The Korean Gastric Cancer Association (www.kgca-i.or.kr)

- Oesophago-Gastric Surgery Section of the Asociación Española de Cirujanos–Spain (www.aecirujanos.es)

- Upper GI International Robotic Association (www.ugira.org)

- United Kingdom Oncology Nursing Society (www.ukons.org.uk)

- The European Oncology Nursing Society (www.cancernurse.eu)

- The Oesophageal Patients Association–United Kingdom (www.opa.org.uk)

- My Gut Feeling–Canada (www.mygutfeeling.ca)

- No Stomach for Cancer–USA (www.nostomachforcancer.org)

- Vivere Senza Stomaco—Italy

- Gastro/Oesophageal Support and Help Cancer Group (Bristol)–United Kingdom

**GASTROS International Working Group Collaborators (To be cited as collaborators in PUBMED)**

- Ademola Adeyeye

- Paulo Matos Costa

- Ismael Diez del Val

- Suzanne Gisbertz

- Ali Guner

- Simon Law

- Hyuk-Joon Lee

- Ziyu Li

- Koji Nakada
- Daniel Reim
- Gian Luca Baiochhi
- William Allum
- Asif Chaudhry
- Ewen Griffiths
- Shuangxi Li
- Yu-long He
- Zekuan Xu
- Yingwei Xue
- Han Liang
- Guoxin Li
- Enhao Zhao
- Philipp Neumann
- Linda O'Neill
- Emer Guinan
- Daniela Zanotti
- Giovanni de Manzoni
- Eliza R.C. Hagens
- Mark I. van Berge Henegouwen
- Patrícia Lages
- Susana Onofre
- Rafael Mauricio Restrepo Nuñez
- Gabriel Salcedo Cabañas
- Maria Posada Gonzalez
- Cristina Marin Campos
- Bahar Candas
- Bahadır Emre Baki
- Muhammed Selim Bodur
- Reyyan Yildirim
- Arif Burak Cekic

## Author Contributions

**Conceptualization:** Bilal Alkhaffaf, Jane M. Blazeby.

**Data curation:** Bilal Alkhaffaf, Aleksandra Metryka.

**Formal analysis:** Bilal Alkhaffaf, Aleksandra Metryka.

**Funding acquisition:** Bilal Alkhaffaf, Jane M. Blazeby, Anne-Marie Glenny, Paula R. Williamson, Iain A. Bruce.

**Investigation:** Bilal Alkhaffaf.

**Methodology:** Bilal Alkhaffaf, Jane M. Blazeby, Paula R. Williamson.

**Project administration:** Bilal Alkhaffaf, Aleksandra Metryka.

**Resources:** Bilal Alkhaffaf.

**Supervision:** Jane M. Blazeby, Anne-Marie Glenny, Paula R. Williamson, Iain A. Bruce.

**Visualization:** Bilal Alkhaffaf.

**Writing – original draft:** Bilal Alkhaffaf, Iain A. Bruce.

**Writing – review & editing:** Bilal Alkhaffaf, Aleksandra Metryka, Jane M. Blazeby, Anne-Marie Glenny, Paula R. Williamson, Iain A. Bruce.

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
