## [Decision Letter · Decision Letter 0]

30 Jun 2021

PONE-D-20-40628

Exploring the impact of regional variation on outcome prioritisation in core outcome set development: a case study in the field of gastric cancer surgery

PLOS ONE

Dear Dr. Alkhaffaf,

Thank you for submitting your manuscript to PLOS ONE. After careful consideration, we feel that it has merit but does not fully meet PLOS ONE’s publication criteria as it currently stands. Therefore, we invite you to submit a revised version of the manuscript that addresses the points raised during the review process.

Please revise the presentation of the study findings. Some guidance is reported below.

We look forward to receiving your revised manuscript.

Kind regards,

Sandro Pasquali, M.D., Ph.D.

Academic Editor

PLOS ONE

Additional Editor Comments (if provided):

Overall the manuscript is very interesting as it investigates an underappreciated topic in surgical research. In this regard, authors should consider that most of readers are not familiar with the study topic and need some more guidance through the manuscript.

This is particularly important as the manuscript is well written, study hypothesis, aims and methodology are clear and conclusions robust. Also, the Discussion section stated “Ultimately, a COS will only achieve its stated goals if researchers use it.”thus supporting the dissemination of the results of this study.

Overall, I would request authors to improve their study presentation in order to make it more readable.

Title is difficult to get and there are repetitions. I would consider a simpler title such as “Development of a core outcome set for clinical trials in gastric cancer surgery: the GASTROS study”.

Abstract is not very informative of study findings. The result section should include the outcomes to be prioritized and report overall rates, otherwise it is very difficult to have an idea of what the manuscript analysed.

Result section is hard to get. For instance, outcomes should be grouped according to their main area, for instance: prognosis, post-operative morbidity, quality of life, long-term morbidity. This may make it easier for Authors to provide a more narrative reporting of their result section, which currently is just a reminder to tables, which is hard to follow for readers. Again , please consider taking the reader through the study results rather than ask him/her to look at tables.

Clearly authors want to reach surgeons and oncologists with this paper. How they plan to reach nurses? What about patients? It would be interesting to have authors’ strategy reported in the discussion.

A study limitation is the lack of medical oncologists and radiation oncologists. Can you please explain reasons for considering only surgeons for a disease that is treated multidisciplinary? This issue should be discussed.

In the study conclusion it should be reported that three stakeholders (surgeons, nurses and patients) were considered. There should at last be some examples of included outcomes, for instance those with the highest agreement.

I would appreciate some comment on the generalizability of these findings to other surgical oncology disease. Authors comment that gastric cancer has extensive differences compared to breast cancer. However these findings may be valuable, at least to some extents, for GI cancers. Please comment.

There is a typo in the first line of 4.4 – “is was”.

Journal Requirements:

3. Thank you for including your ethics statement:  "The study was given ethical approval by the North West - Greater Manchester East Research Ethics Committee (18/NW/0347) and governance approvals by Manchester University Hospitals NHS Foundation Trust.".   

Please provide additional details regarding participant consent. In the ethics statement in the Methods and online submission information, please ensure that you have specified (1) whether consent was informed and (2) what type you obtained (for instance, written or verbal, and if verbal, how it was documented and witnessed). 

4. Please include additional information regarding the survey or questionnaire used in the study and ensure that you have provided sufficient details that others could replicate the analyses. For instance, if you developed a questionnaire as part of this study and it is not under a copyright more restrictive than CC-BY, please include a copy, in both the original language and English, as Supporting Information. 

5. In your Methods section, please provide additional information about the participant recruitment method and the demographic details of your participants. Please ensure you have provided sufficient details to replicate the analyses such as: 

- the recruitment date range (month and year) 

- a description of any inclusion/exclusion criteria that were applied to participant recruitment

- a table of relevant demographic details

- a statement as to whether your sample can be considered representative of a larger population

- a description of how participants were recruited. 

6. Please list the name and version of any software package used for statistical analysis, alongside any relevant references. For more information on PLOS ONE's expectations for statistical reporting, please see https://journals.plos.org/plosone/s/submission-guidelines.#loc-statistical-reporting.

7. We note that the grant information you provided in the ‘Funding Information’ and ‘Financial Disclosure’ sections do not match. 

8. We note that you have indicated that data from this study are available upon request. PLOS only allows data to be available upon request if there are legal or ethical restrictions on sharing data publicly. For more information on unacceptable data access restrictions, please see http://journals.plos.org/plosone/s/data-availability#loc-unacceptable-data-access-restrictions. 

9.One of the noted authors is a group or consortium GASTROS International Working Group . In addition to naming the author group, please list the individual authors and affiliations within this group in the acknowledgments section of your manuscript. Please also indicate clearly a lead author for this group along with a contact email address.

10. We note that Figure 1 in your submission contain map image which may be copyrighted. All PLOS content is published under the Creative Commons Attribution License (CC BY 4.0), which means that the manuscript, images, and Supporting Information files will be freely available online, and any third party is permitted to access, download, copy, distribute, and use these materials in any way, even commercially, with proper attribution. For these reasons, we cannot publish previously copyrighted maps or satellite images created using proprietary data, such as Google software (Google Maps, Street View, and Earth). For more information, see our copyright guidelines: http://journals.plos.org/plosone/s/licenses-and-copyright.

11. Please include a copy of Table 1 which you refer to in your text on page 18.

12. We note you have included a table to which you do not refer in the text of your manuscript. Please ensure that you refer to Table 31 and 32 in your text; if accepted, production will need this reference to link the reader to the Table.

Reviewers' comments:

Reviewer's Responses to Questions

**Comments to the Author**

1. Is the manuscript technically sound, and do the data support the conclusions?

Reviewer #1: Yes

2. Has the statistical analysis been performed appropriately and rigorously? 

Reviewer #1: I Don't Know

3. Have the authors made all data underlying the findings in their manuscript fully available?

Reviewer #1: Yes

4. Is the manuscript presented in an intelligible fashion and written in standard English?

Reviewer #1: Yes

5. Review Comments to the Author

Reviewer #1: This is an interesting paper . I admire your efforts at recruitment for this Delphi.

In answer to the journal’s specific questions:

• Further data/evidence of analysis is offered in response to personal request.

• The manuscript is parsimonious, and well written and presented.

• It offers the tabular analysis I would expect to see to support the body of the paper, and the conclusions drawn seem sound in relation to the data presented. The sample sizes (you do point this out) are too small to make definitive conclusions, and there are many confounding factors (you also point this out) that may play a role in the variation you report.

As a reader:

I was initially not persuaded that subgroup analysis would be pragmatic in this context, but I inevitably felt drawn in by the tables and percentages, and was struck by Table 5 and how the results hang together here. You do state the key point that a COS is intended to allow synthesis, and to be useful in international trials – reminding the reader that the heart of COS methodology is the search for consensus rather than divergence. So the intention to analyse subgroups did raise some questions, but I think I’m persuaded that this was worth exploring (it made me think, always a good sign). Generally speaking, I felt that the paper was carefully worded so as not to stray into too much speculation.

Issues that came to mind as I read through (these are very general):

Section 2.3 Data analysis and interpretation - perhaps I'm misinterpreting this, but the first sentence suggests that participants included in the analysis did not complete the whole Delphi survey - in fact, some of them only completed 50% of the survey? It seemed to me that including surveys that were only partially filled in meant that you are not comparing like-with-like across the different outcomes, but I may be missing something here.

I would have appreciated a sentence or two summarising the concrete differences in pathology, treatment and outcome between regions (rather than the statement that there are differences), so that I could do some independent thinking around this. I don’t see why this paper shouldn’t flag up some potential issues around the consensus approach (rather than just reassuring us!), given that the same methodology for COS development is used across a variety of subject areas which are very different in shape. The COS development process is influenced by the nature of the condition being explored, whether you are trying to tie down outcomes for a single intervention or a range of interventions for the same condition, and the breadth of stakeholder roles involved (which is inevitably affected by the two previous issues). I wanted just a little more information about the background so that I could think about these issues without having to look them up in a separate paper.

It’s a benefit to this analysis to have a low number of stakeholder groups, and it’s a given that a study steering group should carefully think about which role perspectives (besides patients) they should include as respondents in a Delphi survey. I felt that the call to consider more finely grained characteristics of potential respondents when you plan recruitment was perhaps optimistic, as this is partly resource-dependent, and capacity in research institutions varies greatly. Not all COS projects would have funding to access demographic information about patients in order to guide recruitment (and adding demographic questions to the survey could be a burden and increase attrition). I’m not saying this isn’t a good idea for recruitment – just in an ideal world. I wonder whether the paper should touch on capacity issues and make some clear calls for investment in this kind of approach for COS studies that are likely to benefit from it, which would in turn feed back to funders of COS projects?

6. PLOS authors have the option to publish the peer review history of their article (what does this mean?). If published, this will include your full peer review and any attached files.

Reviewer #1: No

---

## [Author Response · Author response to Decision Letter 0]

2 Nov 2021

Response to Reviewer:

1. Section 2.3 Data analysis and interpretation - perhaps I'm misinterpreting this, but the first sentence suggests that participants included in the analysis did not complete the whole Delphi survey - in fact, some of them only completed 50% of the survey? It seemed to me that including surveys that were only partially filled in meant that you are not comparing like-with-like across the different outcomes, but I may be missing something here.

The study management group considered whether to include partially completed surveys. The argument against excluding partially completed surveys related to the possibility that valid responses would be excluded. Ultimately, we settled on including surveys which had been at least 50% completed as it was probable that these participants had understood the purpose of the study. Furthermore, some participants who had not fully completed the first Delphi went on to complete the second survey which supported our choice. The number of partially completed studies was small (3%) and most partially completed surveys had prioritised closer to 100% of outcomes rather than the 50% minimum response threshold that was set.

2. I would have appreciated a sentence or two summarising the concrete differences in pathology, treatment and outcome between regions (rather than the statement that there are differences), so that I could do some independent thinking around this. 

As requested, specific examples of regional differences have now been added to the introduction.

3. I don’t see why this paper shouldn’t flag up some potential issues around the consensus approach (rather than just reassuring us!), given that the same methodology for COS development is used across a variety of subject areas which are very different in shape. The COS development process is influenced by the nature of the condition being explored, whether you are trying to tie down outcomes for a single intervention or a range of interventions for the same condition, and the breadth of stakeholder roles involved (which is inevitably affected by the two previous issues). I wanted just a little more information about the background so that I could think about these issues without having to look them up in a separate paper.

A section which discusses the merits and limitations of using this methodological approach has now been added to the last paragraph before the conclusion.

4. It’s a benefit to this analysis to have a low number of stakeholder groups, and it’s a given that a study steering group should carefully think about which role perspectives (besides patients) they should include as respondents in a Delphi survey. I felt that the call to consider more finely grained characteristics of potential respondents when you plan recruitment was perhaps optimistic, as this is partly resource-dependent, and capacity in research institutions varies greatly. Not all COS projects would have funding to access demographic information about patients in order to guide recruitment (and adding demographic questions to the survey could be a burden and increase attrition). I’m not saying this isn’t a good idea for recruitment – just in an ideal world. I wonder whether the paper should touch on capacity issues and make some clear calls for investment in this kind of approach for COS studies that are likely to benefit from it, which would in turn feed back to funders of COS projects?

The reviewer has made an extremely important point about limited resources often available during COS during studies. However, the demographic information that was gathered at the start of the Delphi survey was relatively basic, was self-reported by participants and did not take long to complete. The concern regarding the burden of supplying demographic details did not materialise during our study which achieved a retention rate of 70% between round 1 and round 2. This has been added to the section 4.1 in the discussion. A call and justification to ensure appropriate funding for COS studies has been incorporated in the discussion. 

Responses to Editor’s Comments

1. Title is difficult to get and there are repetitions. I would consider a simpler title such as “Development of a core outcome set for clinical trials in gastric cancer surgery: the GASTROS study”.

This manuscript aims specifically to explore how outcomes are prioritised by stakeholders in different regions. If outcomes are valued differently, then achieving a balance when recruiting participants for the consensus process becomes more important in reducing bias. The study is not the description of the final COS which has been published elsewhere (https://academic.oup.com/bjs/advance-article/doi/10.1093/bjs/znab192/6308782?searchresult=1). As requested, the title has been simplified to reflect this:

‘How are trial outcomes prioritised by stakeholders from different regions? Analysis of an international Delphi survey to develop a core outcome set in gastric cancer surgery.’

2. Abstract is not very informative of study findings. The result section should include the outcomes to be prioritized and report overall rates, otherwise it is very difficult to have an idea of what the manuscript analysed.

There were 56 outcomes to prioritise. This has now been added to the methods section. The outcomes on which consensus was reached to include in the COS have been added to the abstract. For brevity, those which were excluded or where no consensus was reached have not been added to the abstract but can still be found in the main manuscript.

3. Result section is hard to get. For instance, outcomes should be grouped according to their main area, for instance: prognosis, post-operative morbidity, quality of life, long-term morbidity. This may make it easier for Authors to provide a more narrative reporting of their result section, which currently is just a reminder to tables, which is hard to follow for readers. Again , please consider taking the reader through the study results rather than ask him/her to look at tables.

As requested, the results section has been expanded and a narrative description of the results has now been incorporated. A taxonomy used to organise outcomes for Delphi participants has been cited and tables now reflect these ‘core outcome areas’ as recommended by the reviewer.

4. Clearly authors want to reach surgeons and oncologists with this paper. How they plan to reach nurses? What about patients? It would be interesting to have authors’ strategy reported in the discussion.

A statement regarding our dissemination policy has now been added to the discussion section. Updating all stakeholder groups involved in this study is certainly an important consideration which was set out a priori.

5. A study limitation is the lack of medical oncologists and radiation oncologists. Can you please explain reasons for considering only surgeons for a disease that is treated multidisciplinary? This issue should be discussed.

The scope of the study and the reasons why other healthcare providers were not invited to participate in the study has now been added to the limitations section of the discussion. The justification for this has been included in the methods section.

6. In the study conclusion it should be reported that three stakeholders (surgeons, nurses and patients) were considered. There should at last be some examples of included outcomes, for instance those with the highest agreement.

The conclusion has been modified to reflect the recommendation to explicitly describe the stakeholders invited to participate in the study. The outcomes where consensus agreement was reached amongst all stakeholders has also been included.

7. I would appreciate some comment on the generalizability of these findings to other surgical oncology disease. Authors comment that gastric cancer has extensive differences compared to breast cancer. However these findings may be valuable, at least to some extents, for GI cancers. Please comment.

As this is the first time that a group has examined regional variation as a possible variable on how outcomes are prioritised, it is difficult to be able to comment on whether the findings from this study will be applicable to other COS studies – surgical oncology or otherwise. We certainly believe that this requires some further study and we have added a section into the discussion which encourages other developers to collect this type of baseline data during the registration period of the survey.

The differences that we discuss between gastric and breast cancer relate to the way in which stakeholder groups were assembled during the respective COS, as well as the available treatments. Explicit examples have been given. Further work is already underway to identify commonalities and differences between core outcome sets in surgical oncology; this is an extensive project which would be challenging to summarise or include in the present discussion.

8. There is a typo in the first line of 4.4 – “is was”.

This mistake has been corrected.

---

## [Editor Report · Decision Letter 1]

15 Dec 2021

How are trial outcomes prioritised by stakeholders from different regions? Analysis of an international Delphi survey to develop a core outcome set in gastric cancer surgery.

PONE-D-20-40628R1

Dear Dr. Alkhaffaf,

We’re pleased to inform you that your manuscript has been judged scientifically suitable for publication and will be formally accepted for publication once it meets all outstanding technical requirements. Reviewer and editor's comments have been addressed and manuscript improved accordingly. 

Kind regards,

Sandro Pasquali, M.D., Ph.D.

Academic Editor

PLOS ONE

---

## [Editor Report · Acceptance letter]

22 Dec 2021

PONE-D-20-40628R1 

How are trial outcomes prioritised by stakeholders from different regions? Analysis of an international Delphi survey to develop a core outcome set in gastric cancer surgery. 

Dear Dr. Alkhaffaf:

I'm pleased to inform you that your manuscript has been deemed suitable for publication in PLOS ONE. Congratulations! Your manuscript is now with our production department. 

Kind regards, 

on behalf of

Dr. Sandro Pasquali 

Academic Editor

PLOS ONE